# Image Translation by Latent Union of Subspaces for Cross-Domain Plaque Detection

**Yingying Zhu**[1]                                                     YINGYING.ZHU@NIH.GOV
**Daniel C. Elton**[1]                                                  DANIEL.ELTON@NIH.GOV
**Sungwon Lee**[1]                                                      SUNGWON.LEE@NIH.GOV
**Perry J. Pickhardt**[2]                                          PPICKHARDT2@UWHEALTH.ORG
**Ronald M. Summers**[1]                                                 RMS@NIH.GOV

[1] *Imaging Biomarkers and Computer-Aided Diagnosis Laboratory, Radiology and Imaging Sciences, National Institutes of Health Clinical Center, Bethesda, MD 20892, USA* [2] *School of Medicine and Public Health, University of Wisconsin, Madison, WI 53706, USA*

## Abstract

Calcified plaque in the aorta and pelvic arteries is associated with coronary artery calcification and is a strong predictor of heart attack. Current calcified plaque detection models show poor generalizability to different domains (ie. pre-contrast vs. post-contrast CT scans). Many recent works have shown how cross domain object detection can be improved using an image translation model which translates between domains using a single shared latent space. However, while current image translation models do a good job preserving global/intermediate level structures they often have trouble preserving tiny structures. In medical imaging applications, preserving small structures is important since these structures can carry information which is highly relevant for disease diagnosis. Recent works on image reconstruction show that complex real-world images are better reconstructed using a union of subspaces approach. Since small image patches are used to train the image translation model, it makes sense to enforce that each patch be represented by a linear combination of subspaces which may correspond to the different parts of the body present in that patch. Motivated by this, we propose an image translation network using a shared union of subspaces constraint and show our approach preserves subtle structures (plaques) better than the conventional method. We further applied our method to a cross domain plaque detection task and show significant improvement compared to the state-of-the art method.

## 1. Introduction

Calcified plaques in the aorta and pelvic arteries are associated with coronary artery calcification and are a strong predictor of a heart attack (Parab et al., 2019). Current deep learning models for calcified plaque detection/segmentation require a large amount of labeled training data and show poor generalization ability to different domains. For instance, a segmentation model trained on pre-contrast CT scans will show poor performance on post-contrast CT scans due to the data distribution shift between two domains. To address the domain shift problem, many works show success using image translation models to transfer images to different domains (for sample, pre-contrast CT scans to post-contrast scans) (Zhu et al., 2017; Liu et al., 2017; Isola et al., 2016). One weakness of current image

translation works is that while global and intermediate structures are preserved in synthetic images, subtle structures are often mixed with neighboring larger structures (As shown in Fig.1 (center) and (left)). A common idea used in these previous image translation networks is that images from different domains share a latent single subspace and one can transfer between domains using this domain invariant latent space. These works constrained the image to lie in a single latent space without any constraints on preserving local fine structures. Therefore, it did not preserve the calcified plaque structures after images were transfered to different domains.

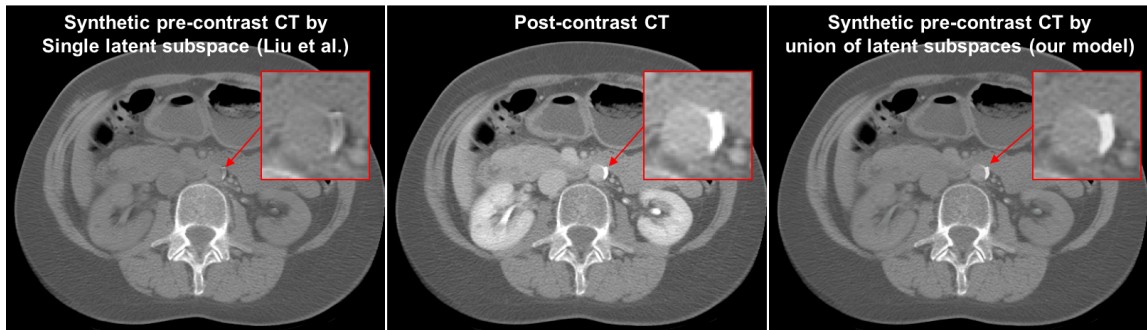

Figure 1: (left) Synthetic pre-contrast CT using shared single latent space model by (Liu et al., 2017). (center) real post-contrast CT scan. (right) synthetic pre-contrast CT using shared latent union of subspace model.

Inspired by recent work has demonstrated that a union of subspaces can improve image reconstruction/generation on natural images especially when it comes to preserving fine structures (Shen et al., 2019; Liang et al., 2018; Zhang et al., 2019; Zhou et al., 2018). We proposed patches based model in this work. We constrained that these extracted image patches to lie in different local subspaces and the whole image lies in a union of these local subspaces. Intuitively these subspaces may be particular organs, bones, and other body parts which are in the patch or of which the patch is a part. We propose to modify the image translation network of (Liu et al., 2017) using a self-expressiveness loss, which is broadly used to model the union of subspace (Ji et al., 2017). We show that the patch based union of the subspace models can preserve subtle structures (such as calcified plaque) in cross-domain image translation compared to state-of-art methods (shown in Fig. 1 right). We further applied this model to assist in a cross-domain calcified plaque segmentation task using a Mask-RCNN based model by (Liu et al., 2019b). Our model shows significant performance improvement compared to the state-of-the-art UNIT method (Liu et al., 2017). It is worth noting that our approach can be generalized to different image translation networks and other types of images besides CT scans.

## 2. Method

Liu et al. proposed (Liu et al., 2017) that the two images $(\mathbf{X}_1, \mathbf{X}_2)$ (paired or unimpaired) from different domains (for example, day or night photos, pre-contrast or post contrast CT

| Testing | Real Pre | Real Post | Liu et al. (60). | Liu et al.(140) | Ours (60) | Ours (140) |
|---------|----------|-----------|------------------|------------------|-----------|------------|
| Precision | 78.4% | 48.6% | 58.5% | 63.2% | **75.7%** | **77.5%** |
| Recall | 82.4% | 54.3% | 64.6% | 69.5% | **78.5%** | **80.7%** |

Table 1: Plaque detection results. The first column gives detection results for the original data without image translation. The 2nd and 3rd columns give results for post-contrast plaque detection after post-contrast to pre-contrast image translation. The difference between training the UNIT image translation model with 60 and 140 scans is also shown.

scans) can be recovered through a shared latent space. If we denote the image encoding functions $\mathbf{E}_1, \mathbf{E}_2$ and imaging decoding/generating functions $\mathbf{G}_1, \mathbf{G}_2$ for the two image domains, then $\mathbf{Z} = \mathbf{E}_1(\mathbf{X}_1) = \mathbf{E}_2(\mathbf{X}_2)$ and $\mathbf{X}_1 = \mathbf{G}_1(\mathbf{Z}), \mathbf{X}_2 = \mathbf{G}_2(\mathbf{Z})$. Here $\mathbf{Z} \in \mathbb{R}^{d \times N}$, where $d$ is the encoded image feature size and $N$ is the number of extracted image patches. Our training image size is $512 \times 512$, and we use cross-validation to find the optimal patch size to be $32 \times 32$. The imaging encoder $\mathbf{E}_1, \mathbf{E}_2$ and decoder $\mathbf{G}_1, \mathbf{G}_2$ from two different domains are trained using an adversarial loss and cycle-consistency constraint to handle the unpaired training data.

Liu et al.'s work shows impressive performance on preserving global structures during image transfer, but we found there was too much loss of detailed information for our application. To force the latent union of subspace structure, we use a self-expressiveness constraint which is satisfied via the following optimization problem:

$$\arg \min_{\mathbf{Z},\mathbf{C},\mathbf{E}_1,\mathbf{E}_2,\mathbf{G}_1,\mathbf{G}_2} \|\mathbf{C}\|_1 + \lambda\|\mathbf{C}\|_*, \tag{1}$$
$$\text{s. t. } \mathbf{Z} = \mathbf{E}_1(\mathbf{X}_1), \mathbf{Z} = \mathbf{E}_2(\mathbf{X}_2), \mathbf{X}_1 = \mathbf{G}_1(\mathbf{Z}), \mathbf{X}_2 = \mathbf{G}_2(\mathbf{Z}), \mathbf{Z} = \mathbf{Z}\mathbf{C}, \text{diag}(\mathbf{C}) = 0$$

Here $\mathbf{C} \in \mathbb{R}^{N \times N}$ is the matrix of self-similarity coefficients. The intuition behind this loss is that different local subspaces are easily separable (sparse similarity coefficients) and well grouped in each local subspace (low rank similarity coefficients). We therefore enforce that $\mathbf{C}$ be both sparse and low rank using the convex surrogate $L_1$ norm and nuclear norm respectively. $\lambda$ is a parameter to balance the trade off between the sparsity and low rank constraints and is tuned using the validation dataset. Further mathematical details can be found in (Zhu et al., 2019; Sui et al., 2019). We implemented the self-similarity constraint using the self-expressiveness network in (Ji et al., 2017; Zhang et al., 2019).

## 3. Results and Discussion

We trained the image translation network using 140 unpaired CT scans (70 pre-contrast, 70 post-contrast) taken from renal donors patients at the University of Wisconsin Medical Center. We trained the plaque segmentation model (Liu et al., 2019b,a) on 75 low dose CT scans which contain a total of 25,200 images, including 2119 with plaques. The test dataset is 30 post-contrast scans and 30 pre-contrast scans from a different dataset, which had plaque labeled manually (7/30 of these scans contained aortic plaques, with a total of 53 plaques overall).

The plaque detection results are shown in Table 1. We trained the image translation networks using different sized training data (60 or 140 scans), for simplicity we discuss the results for 140 scans here. Our model achieved similar plaque detection performance to the real pre-contrast CT scans (precision decreased about 0.9% and recall decreased about 1.7%). Liu et al.'s method, by contrast, shows a 15% drop in precision and a 13% drop in recall caused by loss of fine structures. Interestingly, we obtain similar improvement even when the number of training examples is cut from 140 to 60, showing that this method can be applied even when the number of training data is low. Future work may explore the application of this approach to other domains (such as different contrast phases) and cross domain disease diagnosis.

## 4. Acknowledgments

This research was supported by the Intramural Research Programs of the National Institutes of Health (NIH) Clinical Center and National Library of Medicine (NLM). We thank NVIDIA for GPU card donations.

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
