# OpenReview forum: "Image Translation by Latent Union of Subspaces for Cross-Domain Plaque Segmentation"
_MIDL.io/2020/Conference — MIDL 2020_

### Official Review · AnonReviewer1 · 2020-03-13
**Interesting idea, good experiments**

**Rating:** 4
**Confidence:** 5

**Review:**

The authors propose to extend the UNIT model for image-to-image translation and apply this to synthesis of non-contrast CT images from contrast-enhanced CT images. Subsequent experiments show that aortic calcifications can be automatically identified in the synthetic non-contrast images, but not in the original contrast images.

Strengths
-	It’s very strong that the authors not only perform image-to-image translation, but also evaluate the effect of this translation on a subsequent segmentation task.
-	Good experiments, comparing to a situation without image translation, and one with single subspace image translation. Quantitative results are convincing.
-	Results suggest that the proposed approach would allow automatic aortic calcium scoring in contrast-enhanced images without the need for annotated training data in these images.

Weaknesses
-	Quite information dense paper, it’s not entirely clear what was exactly the contribution of the authors is, e.g. subspace clustering seems to have been proposed previously, as has the UNIT model.
-	The individual models could have been explained a bit better, a diagram would have been useful.
-	It’s unclear what the contribution of the small patches is, it would be interesting to visualize the subspaces using these small patches. Moreover, it’s unclear how the number of patches N is determined.
-	Some typos: ‘for simplicty’, ‘one or more subspace’.

---

### Official Review · AnonReviewer2 · 2020-03-13
**Interesting approach, but very superficial**

**Rating:** 3
**Confidence:** 4

**Review:**

Summary:
This short paper discusses a method for cross-domain plaque detection using image translation methods, translating from pre-contrast to post-constrast CT. The method adds an additional constraint to the learning objective to force the translation model to learn a representation constructed of easily separable subspaces. The authors suggest that this allows the model to better represent the different structures in the images. Their experiments show a decreased drop in performance over competing methods.

Strengths:
This is an interesting application of self-expressiveness loss and the reported results show that the proposed method might achieve a reasonable cross-domain performance. The experiments are very limited, but seem promising.

Weaknesses:
The paper is not very generous with information about the implementation of the method: we are told nothing about the encoding and decoding networks or the segmentation model, for example. The comparison with competing methods is very limited.

The paper seems to depends heavily on work by Liu et al.

It is, of course, a short paper.

Questions:
The main assumption of the authors is that different anatomical structures would be represented by different subspaces in the representation. It would be interesting to know if this is really what happens in the model, or whether the proposed method improves in other ways.

---

### Official Review · AnonReviewer4 · 2020-03-13

**Rating:** 3
**Confidence:** 4

**Review:**

This paper proposes to add a self-expressiveness regularization term to learn a union of subspaces for image-to-image translation in medical domain. It's shown that such self-expressiveness constraint can help to preserve subtle structures during image translation, which is critical for medical tasks, such as plaque detection.

The motivation and methodology are well explained with proper reference works. Improvement on plaque detection is signification.

Comment:
It would nice if the authors could also show some visualisations of the latent space, with comparisons between with and without the constraint. This will provide more insights or explanations.

---

### Official Review · AnonReviewer3 · 2020-03-13
**Improved image translation using UNIT and self-expressive loss for plaque detection.**

**Rating:** 4
**Confidence:** 4

**Review:**

This paper aims to improve image translation for the task of plague segmentation, by building an approach which can preserve small/ subtle structures as well as global and intermediate structures. The approach makes use of UNIT (Liu et al., 2017), an image translation network with a single shared latent space, and a self-expressive loss (Ji et al., 2017), which helps with separating the subspaces and cross-domain translation.
Pros:
The authors explain the task and motivations well, and validate against another well known translation network, UNIT.
The results are promising, with improved performance compared to UNIT. A small decrease in performance is shown in comparison to training on real images, which is to be expected.

Cons:
The method can be more thoroughly validated, and a more detailed illustration of the network can be given.

---

### Meta-Review · Area_Chair1 · 2020-04-05
**MetaReview of Paper266 by AreaChair1**

**Rating:** 4

**Metareview:**

I agree with the reviewers that the paper is quite densely written, it is difficult to understand the details and what exactly the contribution of this work is, but I like that the paper presents image synthesis, which is evaluated with a clinically relevant application. This is an interesting approach for detection of arterial calcifications.

**Paper Type:**

both

---

### Decision · Program_Chairs · 2020-04-11

Accept